# Preclinical validation of an *Escherichia coli* O-antigen glycoconjugate for the prevention of serotype O1 invasive disease

Laurent Chorro,[1] Duston Ndreu,[1] Axay Patel,[1] Srinivas Kodali,[1] Zhenghui Li,[1] David Keeney,[1] Kaushik Dutta,[1] Aniruddha Sasmal,[1] Arthur Illenberger,[1] C. Lynn Torres,[1] Rosalind Pan,[1] Natalie C. Silmon de Monerri,[1] Ling Chu,[1] Raphael Simon,[1] Annaliesa S. Anderson,[1] Robert G. K. Donald[1]

**ABSTRACT** A US collection of invasive *Escherichia coli* serotype O1 bloodstream infection (BSI) isolates were assessed for genotypic and phenotypic diversity as the basis for designing a broadly protective O-antigen vaccine. Eighty percent of the BSI isolate serotype O1 strains were genotypically ST95 O1:K1:H7. The carbohydrate repeat unit structure of the O1a subtype was conserved in the three strains tested representing core genome multi-locus sequence types (MLST) sequence types ST95, ST38, and ST59. A long-chain O1a $CRM_{197}$ lattice glycoconjugate antigen was generated using oxidized polysaccharide and reductive amination chemistry. Two ST95 strains were investigated for use in opsonophagocytic assays (OPA) with immune sera from vaccinated animals and in murine lethal challenge models. Both strains were susceptible to OPA killing with O1a glycoconjugate post-immune sera. One of these, a neonatal sepsis strain, was found to be highly lethal in the murine challenge model for which virulence was shown to be dependent on the presence of the K1 capsule. Mice immunized with the O1a glycoconjugate were protected from challenges with this strain or a second, genotypically related, and similarly virulent neonatal isolate. This long-chain O1a $CRM_{197}$ lattice glycoconjugate shows promise as a component of a multi-valent vaccine to prevent invasive *E. coli* infections.

**IMPORTANCE** The *Escherichia coli* serotype O1 O-antigen serogroup is a common cause of invasive bloodstream infections (BSI) in populations at risk such as newborns and the elderly. Sequencing of US BSI isolates and structural analysis of O polysaccharide antigens purified from strains that are representative of genotypic sub-groups confirmed the relevance of the O1a subtype as a vaccine antigen. O polysaccharide was purified from a strain engineered to produce long-chain O1a O-antigen and was chemically conjugated to $CRM_{197}$ carrier protein. The resulting glycoconjugate elicited functional antibodies and was protective in mice against lethal challenges with virulent K1-encapsulated O1a isolates.

**KEYWORDS** O-antigen, *Escherichia coli*, glycoconjugate vaccine, K1 capsule

Address correspondence to Robert G. K. Donald, Robert.donald@pfizer.com.

All authors were Pfizer employees during the study and may be shareholders of the company. R.G.K.D., A.S.A., L.C., S.K., R.P., and L.C. are inventors on a related patent.

See the funding table on p. 13.

Extraintestinal pathogenic *Escherichia coli* (ExPEC) strains are a leading cause of bloodstream infections (BSI) in high-risk groups such as newborns and the elderly (1–3). In the elderly, urosepsis is the most common clinical syndrome in which the urinary tract is the primary source of infection (4). Invasive uro-pathogenic ExPEC share common surface antigens and virulence determinants, including O-antigens which confer resistance to serum complement (5). Although there are more than 180 different O-serotypes (6, 7), only a limited number of O-antigen serotypes have been associated

with invasive disease (8–10). With increasing resistance to antibiotics, vaccines targeting conserved surface O-antigens offer an alternative means of preventing invasive *E. coli* infections in these vulnerable populations (11). In pioneering work conducted in the early 1990s, an exploratory *E. coli* O-antigen-based glycoconjugate vaccine representing the 12 most prevalent invasive serotypes conjugated to *Pseudomonas aeruginosa* exotoxin A was investigated. These vaccine-elicited antibodies in rabbits were protective against 9 of 12 O-antigen serotypes following IgG passive transfer and lethal challenge in mice but did not confer protection following active immunization in the murine model (12). While well-tolerated in humans (13), the vaccine was not advanced, purportedly due to the availability of effective antibiotic therapies at the time (14). However, variable immune responses to individual O-antigen components and limited evidence of preclinical efficacy for some serotypes may have been additional contributing factors. More recently, a multivalent O-antigen vaccine based on a bioconjugation platform has progressed in clinical trials (NCT04899336) (15, 16), although evidence of efficacy in preclinical models is lacking. Recognizing that configuration of the polysaccharide component of a glycoconjugate can influence immunogenicity, a platform based on high molecular mass $CRM_{197}$ lattice conjugates of genetically modified long-chain O-antigens was optimized and validated in mice with serotype O25b as a test case (10).

Understanding the distribution of O-antigen serotypes is important for estimating vaccine coverage. The O1 serotype ranks third in relative prevalence among contemporary US *E. coli* BSI isolates or fourth among BSI isolates globally (9, 10). Invasive serotype O1 isolates frequently express the K1 capsule, a virulence factor linked to neonatal meningitis, bacteremia, and septicemia (17, 18). The most common cause of Gram-negative bacterial neonatal meningitis is the *E. coli* K1 phylogenetic group of ST95 strains, which are comprised predominantly of O-antigen serotype O1 (19, 20) but also serotypes O18:K1:H7 and O45:K1:H7 (21). In the context of serotype O18, studies with isogenic strains have shown that the K1 capsule is the critical determinant in the development of invasive *E. coli* infections in both neonatal and adult rat models (22, 23), and provides a survival advantage to serotype O1, O6, and O18 isolates in whole blood (5). The *E. coli* sialic acid K1 capsule is structurally equivalent to the *Neisseria meningitidis* serogroup B sialic acid capsule and plays a similar role in immune evasion as an auto-antigen mimic of sialylated host cell glycans such as neural cell adhesion molecule (24–26). Consequently, polysialic acid capsular antigens have not been pursued as vaccine candidates over concerns that antibodies might cause CNS damage. Aside from the K1 polysialic acid capsule, a second virulence factor associated with invasive neonatal O1:K1:H7 strains is a 134 kb-plasmid related to pS88 (20), originally isolated from an O45:K1:H7 meningitis clone (27). This plasmid encodes colicins V and Ia, three iron or manganese capture systems (aerobactin, salmochelin, and *sit*ACBD), and other putative extraintestinal virulence factors, and was shown to play an important role in the pathogenicity of O45:K1 strains in a neonatal rat infection model (27).

In this report, we describe the genotypic diversity of an unbiased sampling of 41 US serotype O1 BSI strains. Virulent strains expressing K1 capsule and harboring pS88 plasmids were used to evaluate the ability of an O1a glycoconjugate antigen to protect mice against lethal challenge. This $CRM_{197}$ lattice glycoconjugate was protective, providing the first evidence that an O-antigen vaccine can prevent disease caused by highly invasive encapsulated isolates of the *E. coli* O1 serotype.

## RESULTS

### Surveillance and genotypic characterization of serotype O1 blood isolates

We previously sequenced 444 invasive *E. coli* bloodstream infection isolates (BSI) collected from 2013 to 2016 to gain insight into the genotypic distribution of serotypes, drug resistance, and virulence determinants (10). From this collection, 41 serotype O1 isolates were identified. The distribution of core genome multi-locus sequence types (MLSTs), *in silico* serotypes, and phylogenies based on whole genome sequencing is shown in Fig. 1; Table S1. Eighty percent of the isolates belonged to ST95 (33/41), 10%

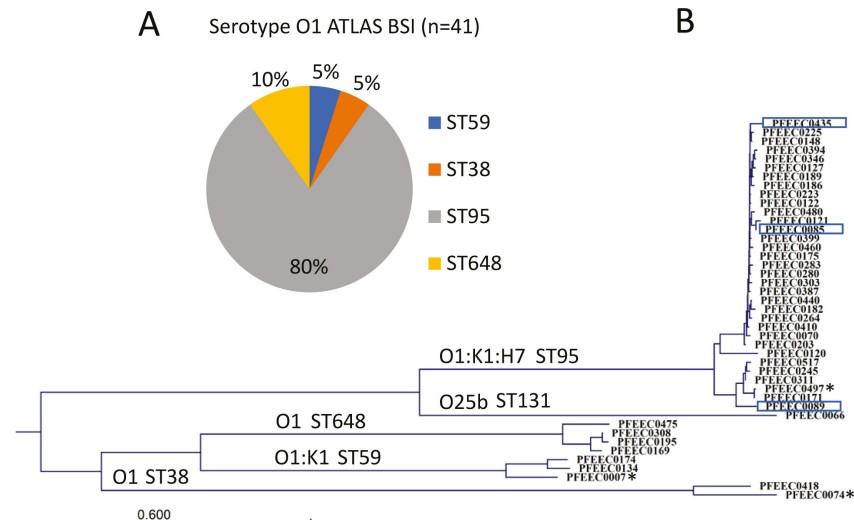

**FIG 1** Genotypic characterization of 41 invasive serotype O1 Antimicrobial Testing Leadership and Surveillance (ATLAS) BSI isolates. (A) Sequence type (ST) distribution, (B) phylogenomic analysis of the *E. coli* O1 blood isolates. The phylogenetic tree of the O1 isolates was constructed using the average nucleotide identity calculated from whole genome sequence alignment using the Neighbor-Joining method. Each branch represents an isolate. The ST of each phylogenetic group is labeled on each branch. The O25b:K5 ST131 isolate PFEEC0066 is included as an outgroup control. ST59 isolate PFEEC0007 is a UTI isolate used for long-chain O-antigen production (see Materials and Methods). Asterisks highlight isolates from which the serotype O1a O-antigen structure was confirmed by nuclear magnetic resonance (NMR) spectroscopy (Fig. 2; Fig. S2 to S4). Blue boxes highlight isolates investigated for their susceptibility to immune serum *in vitro* and/or for their virulence *in vivo*.

were ST648 (4/41), and 5% each (2/41) were ST38 and ST59. All (33/33) of the ST95 isolates formed a distinct genotypic grouping and harbored the biosynthetic *neuDBACES* gene cluster responsible for producing the sialic acid K1 capsule. Serotype O1 BSI strains classified as ST38 ($n = 2$), ST648 ($n = 4$), and ST59 ($n = 2$) formed separate clusters on the phylogenetic tree.

The presence of unique genes within the *waa*Q operon responsible for the biosynthesis of each outer core oligosaccharide can be used to predict lipopolysaccharide (LPS) outer core type (29). Accordingly, the presence of *waa*V was used to predict the R1 core, *waa*K for R2, *waa*D for R3, *waa*X for R4, and *waa*U for K12. Consequently, we determined that the serotype O1 O-antigen is covalently linked to the R4 LPS oligosaccharide core for ST648 and ST38 isolates, all of which lack the K1 sialic acid biosynthetic gene cluster; and is linked to the R1 LPS oligosaccharide core in the K1-encapsulated ST95 and ST59 isolates.

Antibiotic drug resistance was found to be rare among these serotype O1 BSI isolates (Table S1). Only four fluoroquinolone-resistant isolates and four extended-spectrum beta-lactamase strains resistant to third-generation cephalosporins were identified; three of the latter were resistant to both classes of antibiotic but were not associated with a particular genotypic subgroup (ST95, PFEEC0480; ST38, PFEEC0418; ST 648, PFEEC0195). In contrast, analogous rates of resistance to fluoroquinolone or third-generation cephalosporin antibiotics for *E. coli* serotype O25b ST131 BSI isolates in this Antimicrobial Testing Leadership and Surveillance (ATLAS) BSI collection were 93% and 33%, respectively (10).

Eight of the 41 serotype O1 strains (20%) contained a contiguous region of virulence genes indicative of the presence of the pS88-like virulence plasmid. These plasmid-associated virulence factor genes share >98% nucleotide identity with *etsA-C* (putative type 1 secretion system), *arlC (OmpTp*-like protease)*,* and *hylF* (hemolysin); and include biosynthetic gene clusters associated with iron or manganese uptake [aerobactin

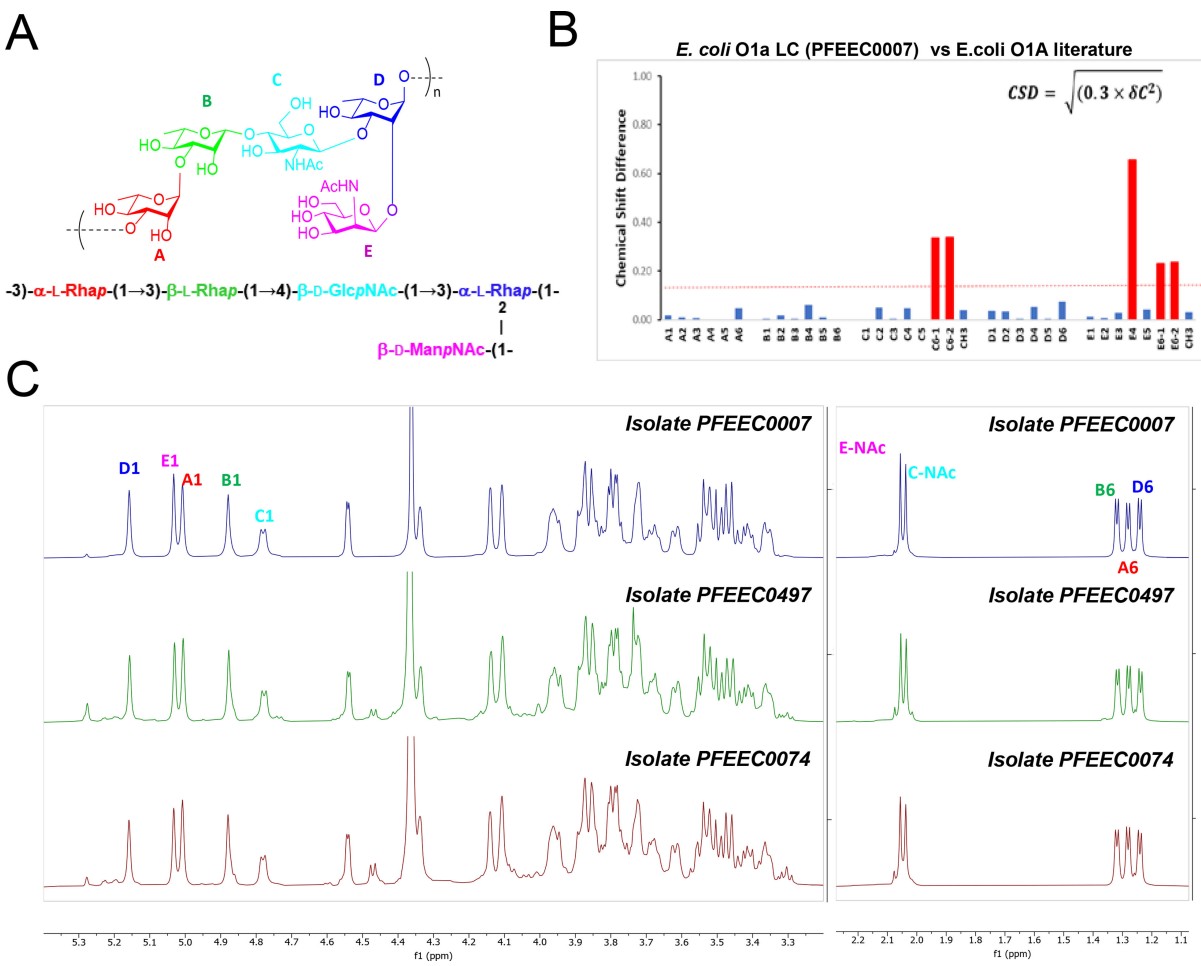

**FIG 2** Structural analysis of serotype O1 O-antigens purified from three BSI isolates. (A) Structure of the *E. coli* O1a polysaccharide antigen repeating unit consisting of five monosaccharide units. (B) The chemical difference plot between O1a long-chain (LC) polysaccharide purified from strain PFEEC0007 (ST59) *wzzB/fepE* and reference values previously reported (28) reveals a few miss-assignments of NMR signals. (C) $^1$H-NMR spectra of *E. coli* O-antigens extracted from blood isolates PFEEC0074 (ST38) (brown), PFEEC0497 (ST95) (green), and PFEEC0007 (ST59) *wzzB/fepE* (blue). The left panel shows the anomeric and ring resonances and the right panel shows the methyl resonances. Some resonances are annotated.

(*iucABCD* and *iutA*), salmochelin (*iroBCDEN*), and the *sit* operon (*sitABCD*)]. Sequencing of plasmid DNA recovered from ST95 O1:K1:H7 strain PFEEC0085 identified a 126.35 kb pS88-like plasmid; 98% of Illumina Miseq DNA fragment raw data reads aligned with the 133.85 kb pS88 reference genome (accession CU928146) (Fig. S1). A pS88-related plasmid was similarly recovered and sequenced from a second ST95 O1:K1:H7 neonatal isolate PFEEC0089. This 133.62 kb plasmid shared 98.3% nucleotide identity with the reference pS88 genome, and except for the introduced chloramphenicol marker, differed only in the absence of two 0.6 kb IS1 elements.

## Characterization of serotype O1 blood isolate O-antigens

To confirm the structural uniformity of the O-antigen oligosaccharide repeat unit (RU), serotype O1 polysaccharides were purified from strains representing distinct ST95 (PFEEC0497), ST38 (PFEEC0074), and ST59 (PFEEC0007) clades. The PFEEC0007 strain was engineered to express a longer chain O-antigen for glycoconjugate production by complementation of a *wzzB* chain length regulator knock-out with a plasmid-borne *Salmonella fepE* gene (10, 30). Detailed structural analysis of the native long- and short-chain polysaccharides produced from the above-mentioned strains was done using 1D and 2D NMR spectroscopy and compared with the published structure.

Structures of the native short-chain and engineered long-chain O-antigens representing the three ST clades matched that of the O1a subtype with correct linkages and sugars, previously associated with O1:K1 invasive strains (28) (Fig. 2A). The resonances assignment in this study was compared with the published assignment (28) by plotting the chemical shift difference between the proton and carbon sugar resonances of O1a polysaccharide (Fig. 2B; Fig. S2). The difference in the resonances observed can be attributed to the missing proton resonances and miss-assignment in the published literature due to the lack of spectral resolution and assignment done using homonuclear $^1H$-$^1H$ NMR experiments. The corrected proton and carbon assignments of O1a polysaccharide repeating unit are shown in Fig. S2. Analysis of 1D $^1H$ spectra (Fig. 2C; Fig. S3) and 2D $^1H$-$^{13}C$ HSQC spectra (Fig. S4) of polysaccharide obtained from PFEEC0007 (ST59), PFEEC0497 (ST95), and PFEEC0074 (ST38) strains confirm that all three strains produce the O1a polysaccharide.

The structure of the O-antigen representing the four ST648 O1 serotype strains was inferred indirectly, through genotypic analysis of their biosynthetic gene clusters. Sequence analysis revealed that the four ST648 strain O-antigen biosynthetic gene clusters were 100% identical to each other across 10.05 kb of operon sequence spanning genes *rmlB* through *wekO* (Fig. S5). Only minor SNP differences were identified with the analogous sequences from the ST59, ST95, and ST38 isolates structurally confirmed as serotype O1a by NMR analysis; conservative amino acid substitutions in *wzx* (L181I), *wekM* (I202V), *wekN* (E264D), and *wekO* (L166V) genes were identified in one or two (but not all three) of these reference isolates. The common operon gene organization and nucleotide level similarity are therefore consistent with a common O1a subtype O-antigen structure for isolates from all four MLST sequence types.

## Construction of the conjugate vaccine candidate

Long-chain O1a polysaccharide purified from ST59 strain PFEEC0007 *wzzB/fepE* was conjugated to $CRM_{197}$ carrier protein to generate hyper-immune rabbit sera for isolate phenotyping and for subsequent mouse immunogenicity and vaccine antigen efficacy studies. Properties of the polysaccharide and derived glycoconjugate are shown in Table 1. The polysaccharide molecular mass, determined by size exclusion chromatography-multi angle laser light scattering (SEC-MALLS), was 50 kDa, which was 3.2–3.9 times the mass of the corresponding native short-chain O-antigens purified from strains PFEEC0074 (15.2 kDa) and PFEEC0497 (12.97 kDa). Long chain and native polysaccharides have approximately 57 and 15–18 RUs, respectively. The longer O-antigen polysaccharide was conjugated to $CRM_{197}$ carrier protein at a saccharide:protein mass ratio (SPR) of 1:1, using reductive amination chemistry in dimethyl sulfoxide (DMSO) solvent (RAC/DMSO) chemistry (10, 31). Endotoxin levels for the unconjugated long-chain polysaccharide or derived antigen conjugate were confirmed to be below 0.1 EU/dose/animal (0.05 EU/µg antigen).

**TABLE 1** O1a O-antigen properties[a]

| O-antigen type | O-antigen | | RAC/DMSO $CRM_{197}$ conjugate |
| | Native | Long chain | Long chain |
| --- | --- | --- | --- |
| Activation | – | – | DO:13 |
| SPR ratio | – | – | 1.0 |
| Free saccharide | – | – | 20% |
| MW (kDa) | 12–15 | 50 | 1,284 |
| Repeat units (RU) | 15–18 | 57 | 57 |

[a]Attributes of the glycoconjugate are described in terms of saccharide to protein mass ratio (SPR), degree of activation of the polysaccharide by periodate (DO), and molecular weight (MW). DO of 13 indicates that 1 RU is modified per every 13 RUs or 7.7%. The number of repeat units per polysaccharide chain was calculated from the ratio of experimental MW (SEC-MALLS) of polysaccharide to the MW of the repeating unit (844.81 Da). Polysaccharides include residual LPS outer core residues which contribute ~2 kDa to the overall MW. '-' not applicable for unconjugated O-antigens.

## Susceptibility of ST95 O1:K1:H7 isolates to O-antigen immune sera

Rabbit antiserum to the glycoconjugate cross-reacted with LPS purified from 85% (35/41) of the ATLAS O1 blood isolates (Table S1; Fig. S6). Of the six negative strains, four expressed truncated LPS that co-migrated on SDS-PAGE gels with control LPS from ST95 strains PFEEC0085 and PFEC0435 harboring $\Delta waaL$ knock-out mutations, that prevent the ligation of O-antigen to the LPS core. LPS from the two remaining strains PFEEC0410 and PFEEC0418 that failed to cross-react with the O1a-antisera had normal LPS SDS-PAGE profiles. For strain PFEEC0418 the lack of cross-reactivity could be explained by the presence of an anomalous chimeric O1 O-antigen biosynthetic gene cluster.

Two K1-encapsulated ST95 isolates PFEEC0085 and PFEEC0435 were subsequently characterized to verify the surface expression of O-antigen and K1 capsular polysaccharides and to investigate their susceptibility to killing by serotype O1a immune sera. Isogenic null mutant strains unable to express K-antigen were generated by deleting genes encoding the the $kpsD$ group II transporter gene. Flow cytometry experiments using the rabbit polyclonal O1a CRM$_{197}$ glycoconjugate immune sera and a polysialic acid-specific monoclonal antibody for antigen detection confirmed that loss of O1 and K1 antigen surface expression in $waaL$ or $kpsD$ mutant strains could be restored by complementation with $waaL$ or $kpsD$ plasmids (Fig. S7).

Next, we tested the sensitivity of both strains to OPA killing, using serotype O1a O-antigen specific rabbit immune sera in the presence of neutrophil-like HL60 effector cells and subinhibitory levels of baby rabbit complement. Immune sera sampled from two rabbits 15 wk after immunization with three 10 µg doses of O1a CRM$_{197}$ glycoconjugate were highly potent in killing each strain (Fig. 3A and B). The bactericidal potency of serially diluted immune serum was assessed relative to a control reaction lacking any added serum antibodies (dotted lines labeled "BG" in Fig. 3A and B). Relative to this control, no OPA killing was observed for either strain with rabbit pre-vaccination serum. Average serum dilution OPA EC$_{50}$ titers at which 50% killing was observed relative to the control for strain PFEEC0085 were 3,840 (rabbit 1) and 3,430 (rabbit 2), and for strain PFEEC0435 the corresponding values were 2,830 (rabbit 1) and 15,160 (rabbit 2).

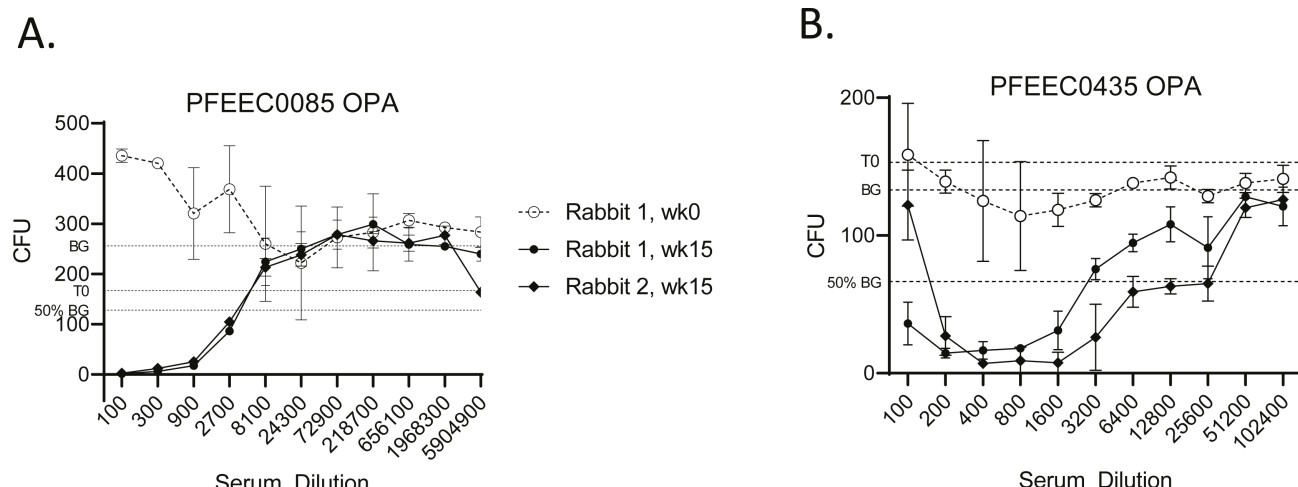

**FIG 3** *E. coli* O1:K1 strains PFEEC0085 (A) and PFEEC0435 (B) are susceptible to OPA killing by rabbit O1a glycoconjugate immune serum in the presence of HL60 effector cells and baby rabbit complement. The activity of immune serum samples after three 10 µg antigen doses with 50 µg AlPO$_4$ adjuvant (week 15 timepoint) was compared to preimmune control sera (week 0 timepoint). T0, input CFUs; BG, background CFUs from no-serum control incubation. 50% BG, half maximal growth inhibition CFU value for determining EC$_{50}$ titers. Plotted data are the average of duplicate titrations with StDev error bars. Optimized OPAs developed for strains PFEEC0085 and PFEEC0435 required bacteria harvested at a mid-log phase in Todd-Hewitt broth (THB) and Dulbecco's modified Eagle's medium (DMEM) media, respectively. Despite comparable input CFUs, final CFUs were higher for strain PFEEC0085 due to growth during the assay incubation period.

## K1 capsule contributes to the virulence of an ST95 O1:K1:H7 strain

To assess suitability for use in vaccine protection studies, the virulence of strains PFEEC0085 and PFEEC0435 was evaluated in immunocompetent CD-1 mice following intraperitoneal (i.p.) challenge. Mice were inoculated i.p. with bacteria at doses ranging from ~5 × 10$^6$ to ~4 × 10$^8$ CFU/dose, and survival of the animals was monitored for 72h. Isogenic unencapsulated *kpsD* null mutant strains were similarly evaluated to assess the role of the K1 capsule in bacterial virulence. Strain PFEEC0085 was substantially more virulent than PFEEC0435, requiring a 35-fold lower i.p. challenge dose (of 3.4 × 10$^6$ versus 1.2 × 10$^8$ CFU/animal) to achieve LD$_{80}$ lethality (Fig. 4A and B; Table 2). In contrast, mice infected with PFEEC0435 did not succumb to disease at an injection dose level of less than 3 × 10$^7$ CFUs. Virulence of PFEEC0085 was highly dependent on the presence of the K1 capsule, as the derived *kpsD* mutant was as mildly virulent as the PFEEC0435 strain. Analogous loss of K1 capsule expression in PFEEC0435 had no significant impact on the intrinsically lower virulence of this strain.

Due to its high degree of virulence, PFEEC0085 was selected as a challenge strain for vaccine efficacy studies described below. A second similarly virulent K1-encapsulated neonatal invasive isolate PFEEC0089 was also selected for this purpose (Table 2). Challenge doses for each strain were selected (LD$_{80}$ i.p.), and calibrated to achieve ~20% survival in the PBS-treated control group, a target threshold chosen for experimental reproducibility and sensitivity.

## Immunogenicity and efficacy of the O1a glycoconjugate in mice

Groups of twenty CD-1 mice were immunized three times with 0.2 µg or 2 µg per dose of either unconjugated O1a polysaccharide or the O1a long-chain CRM$_{197}$ conjugate without adjuvant (Fig. 5A). Serum IgG and OPA titers were assessed after the second and third vaccinations and showed dose-dependent responses (Fig. 5B and C; Table S2). While uniformly strong IgG titers were observed at the 2 µg dose level after two doses, a third 2 µg dose of vaccine was required to elicit functional OPA titers in 90% of the

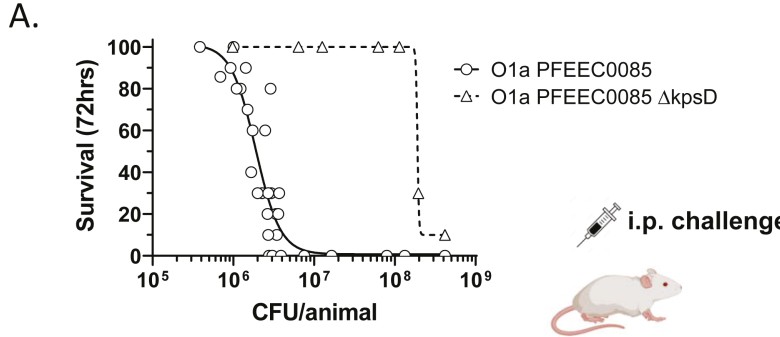

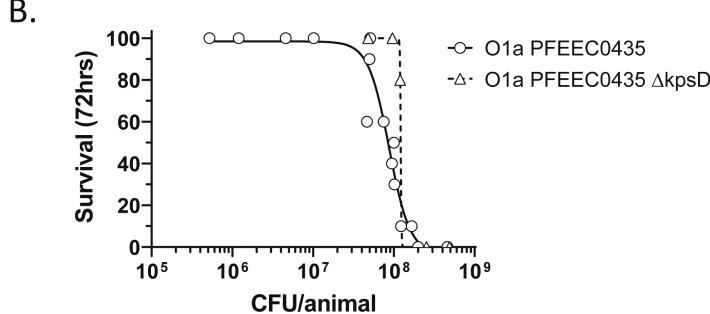

**FIG 4** Distinct virulence properties of two ST95 O1:K1:H7 isolates. (A and B) Susceptibility of strains PFEEC0085 and PFEEC0435 and derived K1 mutants to i.p. challenge. Each symbol represents a group of 10 mice challenged with the indicated inoculum of bacteria (CFU/animal).

**TABLE 2** Properties of O1a K1:H7 ST95 BSI isolates evaluated for virulence in mice

| Isolates | Isolate source | Patient age | pS88 plasmid | K1-CPS[a] | LD$_{80}$ challenge dose i.p. (CFU/animal) |
|---|---|---|---|---|---|
| PFEE0085 | 2013, Nebraska | −1 (neonate) | Yes | Yes | $3.4 \times 10^6$ |
| PFEE0085 Δ*kpsD* | | | Yes | No | $\sim 2 \times 10^8$ |
| PFEEC0435 | 2015, New York | 71 | No | Yes | $1.2 \times 10^8$ |
| PFEEC0435 Δ*kpsD* | | | No | No | $1.2 \times 10^8$ |
| PFEEC0089 | 2013, Nebraska | −1 (neonate) | Yes | Yes | $6.0 \times 10^6$ |

[a]Flow-cytometry with K1-specific mAb 13D9-151 (NRC, Canada).

vaccinated mice. Analogous OPA responses after three 0.2 µg doses resulted in a fivefold lower GMT and a 60% responder rate. In contrast, mice immunized with unconjugated O1a polysaccharides mounted no significant anti-O1a IgG responses, comparable to the baseline levels observed with the PBS control group.

To assess protective efficacy, mice vaccinated with three doses of 0.2 µg or 2 µg of the monovalent O1a glycoconjugate were split into two subgroups of 10 mice each and challenged separately with a lethal dose of either strain PFEEC0085 or strain PFEEC0089. As shown in Fig. 5D, three doses of the CRM$_{197}$ conjugate at 2 µg provided >90%

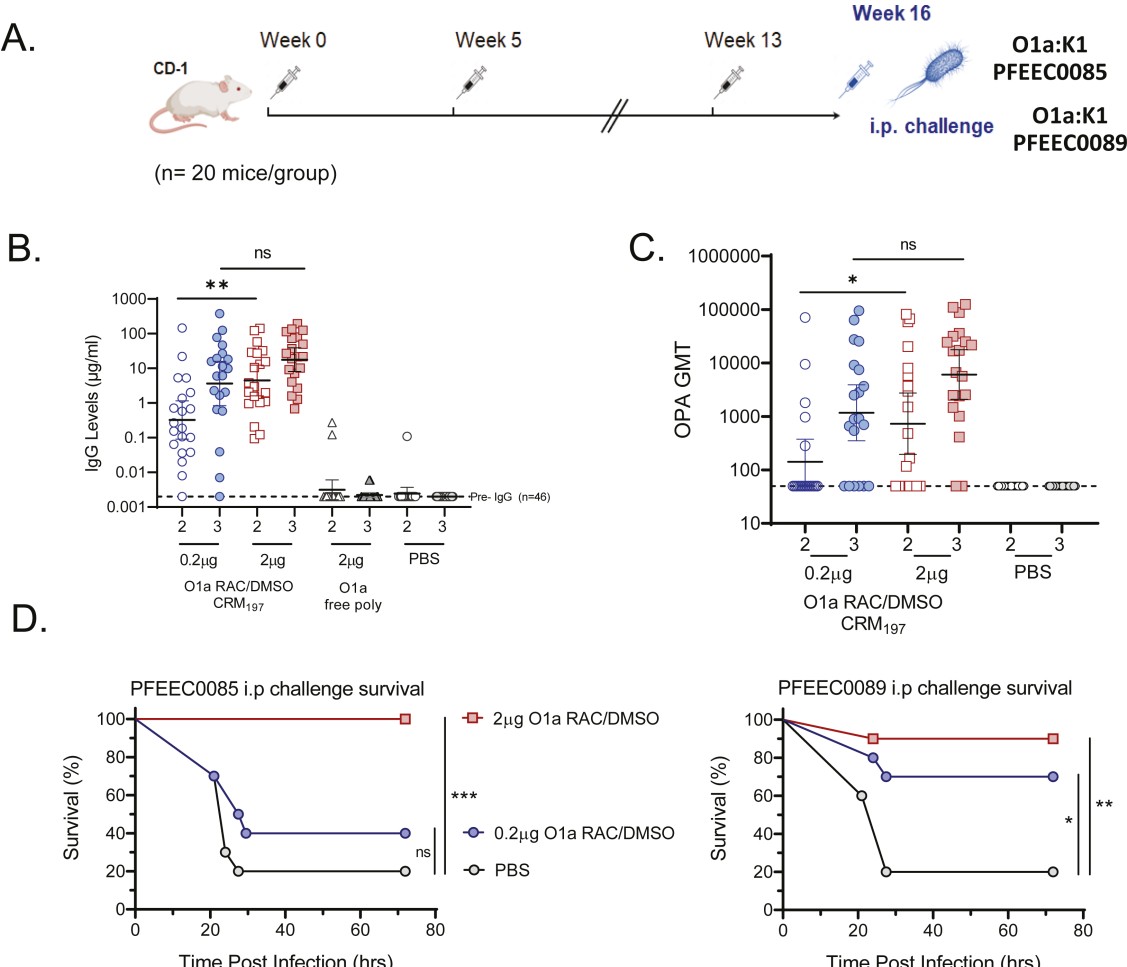

**FIG 5** Immunogenicity and efficacy of the O1a CRM$_{197}$ glycoconjugate. (A) Dosing and challenge schedule (20 mice per group). (B and C) Antigen-specific IgG titers and OPA titers were generated with strain PFEE0085 at week 7 (PD2) and week 15 (PD3) time points. Dotted lines represent GMT titers of mouse naïve serum baseline controls. Corresponding IgG and OPA GMT values are summarized in Table S2. (D) Survival of mice (*n* = 10) following lethal challenge at week 16 with strains PFEEC0085 and PFEEC0089 after three vaccine doses. Asterisks indicate *P* values between bracketed groups. Statistical survival differences were determined using a log-rank test (Mantel-Cox). ***, *P* < 0.001; **, *P* < 0.01; *, *P* < 0.05; ns, not significant.

protection against either virulent strain in this model, whereas 80% of mice in control groups vaccinated with PBS succumbed to infection within 48 h. The 0.2 µg dose was partially (70%) protective against strain PFEEC0089 and only marginally (40%) protective against strain PFEEC0085.

## DISCUSSION

The *E. coli* O1 antigen serogroup is comprised of three structurally distinct subclasses. The O1a subtype is associated with virulent strains (28), while minor subtypes O1b and O1c subtypes have not previously been reported among pathogenic *E. coli* (32). O1b and O1c differ from O1a in their monosaccharide repeat unit composition, in which a β-L rhamnose (residue B in Fig. 2A) is substituted with a 2- or 3-linked α-D galactose (32). We confirmed the O1a structure is present in invasive strains representing three distinct phylogenetic clades of serotype O1 isolates belonging to ST95, ST38, and ST59. Furthermore, 85% of the O1 BSI strains identified by *in silico* serotyping expressed LPS that cross-reacted with antisera to the glycoconjugate of the O1a polysaccharide. Therefore, we believe that these findings support that the O1a O-antigen is the appropriate subtype to include as a component of a multivalent O-antigen vaccine to protect against sepsis in vulnerable patients (10).

Our survey of 41 randomly selected serotype O1 ATLAS BSI isolates confirmed the predominance of the ST95 O1:K1:H7 clonal group, with a prevalence of 80% (33/41). We chose to characterize two ST95 O1:K1:H7 isolates in detail to investigate the contribution of surface polysaccharide antigens and plasmid-encoded factors to virulence and susceptibility to killing by O-antigen immune sera: PFEEC0085, a neonatal sepsis strain containing a pS88 plasmid is lethal to mice at a relatively low challenge dose in an i.p. lethal challenge model ($LD_{80}$ $3 \times 10^6$); and PFEEC00435, a less virulent strain isolated from an elderly bacteremia patient lacking the pS88 plasmid. Both ST95 O1:K1:H7 strains were highly susceptible to killing by O1a glycoconjugate immune sera in the presence of HL60 effector cells and sub-inhibitory levels of serum complement. By comparing the virulence properties of isogenic strain pairs, the K1 capsule was found to be a key contributor to O1:K1:H7 PFEEC0085 strain hypervirulence. Previously the K1 polysialic capsule was similarly shown to be a critical determinant for the ability of an *E. coli* serotype O18:K1 strain to cause meningitis in a newborn rat model (33). The presence of the pS88-like plasmid in PFEEC0085 and the absence of the genotypically similar, but lower virulence ST95 O1:K1:H7 isolate PFEEC0435, suggests a possible role of the episome in potentiating infection. However, our attempts to link the presence of plasmid to the virulence of either PFEEC0085 (by plasmid curing) or other ST95 O1a:K1:H7 isolates (by association) were not successful (data not shown). Therefore, while the enhanced invasiveness of the PFEEC0085 strain is dependent on the K1 capsule, the potential contribution of the pS88-like plasmid remains to be determined.

Regardless of the potential for capsular polysaccharides to block the binding of O-antigen-elicited antibodies *in vitro*, we have demonstrated that $CRM_{197}$ O-antigen lattice glycoconjugates are protective *in vivo*. We showed previously that long-chain O25b glycoconjugates protected against the lethal challenge of CD-1 mice with highly virulent K5 and K100 encapsulated serotype O25b ST131 isolates. In this report, we show that an analogous long-chain O1a lattice glycoconjugate can protect against two similarly virulent K1-encapsulated serotype O1 ST95 isolates. Collectively these results indicate that *E. coli* clinical isolates likely have O-antigens exposed to circulating antibodies during disseminating infection. Studies with passively infused O-antigen monoclonal antibodies indicate that complement-mediated bactericidal or opsonophagocytic killing is the primary mechanism of protection against invasive *E. coli* in various murine or rat challenge models (34, 35). The candidate O25b and O1a long-chain glycoconjugate antigens we have described show promise as components of a multivalent vaccine to confer analogous long-term polyclonal antibody-based prophylaxis against invasive *E. coli* infections.

## MATERIALS AND METHODS

### Strains

An *E. coli* serotype O1 strain from the Wyeth Tygacil collection PFEEC0007 (Genbank accession PRJNA872964) was initially sequenced and modified to produce long-chain O-antigen by introducing a *wzzB* deletion and plasmid expressing the *Salmonella enterica* serovar *Typhimurium fepE* chain length regulator (36). This strain is an ST59 serotype O1 strain isolated in 2001 from a urinary tract infection. Separately, with the goal of surveillance, *E. coli* clinical BSI isolates from US hospitals were randomly selected from the Pfizer-sponsored ATLAS collection, maintained by the International Health Management Associates clinical lab. Strains were genotypically characterized by whole-genome sequencing (WGS) using the MiSeq platform (Illumina) and sequences were deposited in Genbank (accession PRJNA804716) (10). *In silico* genotyping used the Warwick MLST scheme, and O and H typing, LPS oligosaccharide core type, and FimH typing algorithms (29, 37–39). *E. coli waaL* or *kpsD* genes in serotype O1:K1:H7 strains PFEEC0085 and PFEEC0435 were deleted using the λ-Red-mediated homologous recombination system (40), obtained from the Yale University *E. coli* genetic stock center. For genetic complementation of K1 and O-antigen knockout mutations, *kpsD* and *waaL* genes were amplified from strain PFEEC0435 and subcloned using 5′*Nhe*I and 3′*Hind*III adaptor primers into the arabinose-inducible expression vector pBAD18 (41).

### Virulence plasmid isolation and sequencing

A chloramphenicol (CAT) cassette was introduced into an intergenic region of p88 plasmids in strains PFEEC0085 and PFEEC0089 via the lambda red recombinase system (40) and CAT-tagged plasmid DNA purified and transfected into *E. coli* DH10B lab strain by electroporation. Purification of large plasmids from chromosomal DNA contamination was done using a large construct purification kit (Qiagen). DNA from the enriched material was sequenced using the Miseq platform. CLC software was initially used to align raw read data from total genomic DNA from PFEEC0085 to reference pS88 NC_011747, which generated a contig. This contig was used as a guide for the SPADES algorithm-mediated assembly of plasmid raw data to generate a final pS88 contig. Open reading frames were assigned using a combination of Prokka (42) and pS88-reference annotations. Sequences of the pS88-like plasmids from PFEEC0085 and PFEE0089 are deposited in GenBank (accession numbers OP331340 and OP331339). Orientation of an IS-element mediated inversion in the pS88 plasmid from PFEEC0087 was confirmed by PCR using primers flanking the proximal and distal IS1-elements, which yielded the expected 1.5 kb DNA fragments. No evidence of indel or secondary mutations following whole genome sequencing was observed in chromosomal or plasmid sequences following the introduction of the *kpsD* deletion into strain PFEEC0085 to assess the impact on virulence.

### O-antigen purification

Long-chain serotype O1a O-antigen from strain PFEEC0007 *wzzB/fepE* was purified as described previously for the O25b long-chain O-antigen (10). Native short-chain *E. coli* serotype O1 O-antigens were extracted from bacteria harvested from 500 mL shake flask cultures. After washing the cell pellet in water, the pH was adjusted to 3.8 with glacial acetic acid followed by hydrolysis in a 100°C water bath for 2 h. The suspension was cooled and then neutralized with 14% ammonium hydroxide. A solid-liquid separation was performed by centrifugation (9,000 × *g*, 25 min) and the supernatant was collected. Next, the crude O-antigen solution was flocculated using alum solution (2% wt/vol) and pH was adjusted to 3.2 using 1N sulfuric acid. After 1 h of incubation at RT, the suspension was clarified by centrifugation (12,000 × *g*, 35 min, 15°C). Further purification of O-antigen was accomplished by utilizing ultra-filtration/dia-filtration (UFDF). Using an Ultracel 5-kD membrane in a Labscale TFF system, first, the O-antigen solution was reduced to ~40 mL and then diafiltered with 25 mM citrate + 0.1M NaCl buffer (20×

diavolume); a second diafiltration was performed with 25 mM Tris-HCl + 25 mM NaCl buffer (20× diavolume). The UFDF retentate was then purified using anion-exchange membrane chromatography (with 25 mM Tris-HCl + 25 mM NaCl elution buffer). Four molar ammonium sulfate was added to the eluate to a final concentration of 2M. This mixture was purified by hydrophobic interaction chromatography and the elute was collected. Final UFDF (5 kD Ultracel membrane, 30× diavolume of water) purification, extensive dialysis (3.5 kD dialysis cassette, 8 × 4 L water, RT), and final lyophilization yielded pure O-antigen in solid form.

## Structural analysis via NMR spectroscopy

Lyophilized O1a O-antigen polysaccharide was weighed and dissolved in $D_2O$. The resuspended polysaccharide was spun down to remove any insoluble aggregates. The sample was transferred into the 5 mm NMR tube for data collection. All data were collected at 70°C using a Bruker 600 MHz spectrometer equipped with TCI cryoprobe. The following 1D and 2D NMR data were collected for complete structural analysis: 1D $^1H$ (d1 5s; 32 k points), 2D $^1H$-$^{13}C$ HSQC, $^1H$-$^{13}C$ HMBC, $^1H$-$^{13}C$ HSQC-COSY, $^1H$-$^{13}C$ HSQC-TOCSY with mixing time of 120 ms, and multiplicity edited $^1H$-$^{13}C$ HSQC. The 1JCH correlations were determined using the $^1H$-$^{13}C$ CLIP-HSQC experiments. All 2D experiments were carried out using 2,024 and 256 data points in the $^1H$ and $^{13}C$ dimensions, respectively. The NMR data were processed using Topspin or NvX and analyzed using Topspin or NMRViewJ software (43). The chemical shift difference was plotted using this equation:

$$\text{CSD} = \sqrt{\delta H^2 + \left(0.3 \times \delta C^2\right)}$$

where, δH and δC are the proton and carbon chemical shifts, respectively. Due to the missing proton resonances in the literature, the chemical shift difference was plotted using the carbon resonances for the data shown in Fig. 2B. The molar mass of the O-antigen was determined by SEC-MALLS.

## Animal studies

To evaluate immunogenicity, two eight-week-old female New Zealand white rabbits (8 wk old, Charles River Labs) were immunized via intramuscular injection with 10 µg (0.5 mL) of O1a glycoconjugate and 50 µg $AlPO_4$ adjuvant at weeks 0, 6, and 14. Immune sera was sampled at week 15. Six to eight-week-old female CD-1 mice ($n = 20$ per group, Charles River Laboratories) were immunized by subcutaneous injection with 0.1 mL of vehicle control (PBS: phosphate-buffered saline pH 7.0), 0.2 µg or 2 µg of monovalent O1a polysaccharide conjugate, or 2 µg of unconjugated O1a polysaccharide. Immunizations were administered at weeks 0, 5, and 13. All mice were bled via submandibular route and sera were heat-inactivated for 1 h at 60°C. Sera were collected at weeks 0, 7, and 15 post-immunization and prior to the challenge. For the *E. coli* septicemia model, 12 to 14-week-old female CD-1 mice ($n = 10$ mice per group, Charles River Laboratories) were injected intraperitoneally (i.p.) with various doses of O1:K1:H7 isolates, PFEEC0085 or PFEEC0089, to identify an optimal lethal dose with 80% death ($LD_{80}$). All bacterial inoculums were prepared in PBS. To evaluate the efficacy of the O1 glycoconjugate antigen, immunized mice were inoculated i.p. with a lethal dose of $3–6 \times 10^6$ CFU per animal of *E. coli* O1:K1 isolates. Animal survival was monitored every 3 h for 72 h. The remaining animals were humanely exsanguinated at the study endpoint. Mantel-Cox Log-rank statistical analysis was used to compare survival differences between challenged groups. Animals were provided with a standard diet and unrestricted access to water. All animal studies complied with Pfizer's local and global Institutional Animal Care and Use Committee guidelines and the facility is accredited by the International Association for Assessment and Accreditation of Laboratory Animal Care.

## Flow cytometry strain characterization

Surface antigen expression was measured by flow cytometry (iQue Advanced Flow Cytometer, Sartorius). Bacteria were fixed in 4% paraformaldehyde and stained with primary antibodies: 4 µg/mL polysialic acid capsule specific mAb 13D9-151 (NRC, Canada) and 1:100 serum dilutions of rabbit preimmunization sera or O1a glycoconjugate post-vaccination sera. Antigens were detected with phycoerythrin (PE)-conjugated anti-mouse or anti-rabbit secondary antibodies.

## Detection of antigen-specific IgG in sera

For detection of serotype O1a specific IgG, long-chain O-antigen polysaccharide was covalently conjugated to poly-L-lysine with 1-cyano-4-dimethylaminopyridinium, and the derived poly-L-lysine conjugate was covalently coupled to magnetic carboxy bead microspheres (Magplex; Luminex) with EDC/NHS [1-ethyl-3-(3-dimethylamino) propyl carbodiimide/N-hydroxysuccinimide] (Thermo Fisher). Beads were incubated with serially diluted individual mouse sera or control mAb with shaking at 4℃ for 18 h. After washing, bound serotype-specific IgG was detected with a PE-conjugated goat anti-mouse total IgG secondary antibody (Jackson ImmunoResearch) after 60 min of RT incubation. Microplates were read on a FlexMap 3D instrument (Bio-Rad). A serotype-specific IgG mAb was used as an internal standard to quantify IgG levels. A standard curve plot for the mAb titration yielded a linear slope profile across a $10^3$ range of serum dilutions (log fluorescence versus log serum dilution).

## Serum opsonophagocytic assays

The assay is an adaptation of the classical killing-type OPA used for serological evaluation of pneumococcal vaccines (44). Undifferentiated HL60 cells (ATCC CCL-240), maintained in RPMI supplemented with 10% HI-FCII (Fetal clone II sera) were differentiated with 100 mM dimethylformamide. Three or four days post-differentiation, cells were harvested, and viability and phenotypic characteristics were determined by flow cytometry to analyze whether cells passed the acceptance criteria. Differentiated HL-60 cells were considered acceptable as effector cells in the OPA if they met the acceptance criteria of ≥55% CD35+ and ≤15% CD71+. Bacterial stocks were prepared by growing bacteria in Dulbecco's modified Eagle's medium (DMEM) or Todd-Hewitt broth (THB) to an optical density at 600 nm ($OD_{600}$) of between 0.5 and 1.0, and glycerol was added to a final concentration of 20% prior to freezing. Exploratory assays used a 96-well microplate format. Pre-titered thawed bacteria were diluted to $1 \times 10^5$ CFU/mL in OPA buffer [Hanks balanced salt solution (Life Technologies), 0.1% gelatin], and 10 µL ($10^3$ CFU) of the bacterial suspension was combined with 40 µL of buffer and 20 µL of serially diluted serum for 30 min at RT in a 96-well tissue culture microplate. Subsequently, 10 µL of 2% complement (baby rabbit serum Pel-Freez) and 20 µL of HL60 cells (at 200:1 effector cell-to-bacterial-target ratio) were added to each well, and the mixture was shaken at 700 rpm for 45 min at 37℃ in a 5% $CO_2$ incubator. After the incubation, 10 µL of each 100 µL reaction mixture was transferred into the corresponding wells of a prewetted 96-well Millipore MultiScreen HTS HV filter plate containing 150 µL DMEM/well. The liquid was removed by vacuum filtration and the plate incubated overnight at 37℃ in a sealed bag. The next day, the colonies were enumerated after staining with Coomassie dye using an ImmunoSpot analyzer and Immunocapture software (Cellular Technology, Ltd.). To establish the specificity of OPA activity, immune sera were preincubated with 20 µg/mL of the homologous serotype purified O antigen prior to the opsonization step. The OPA included control reactions without HL60 cells or complement, to demonstrate the dependence of any observed killing on these components. Individual serum OPA titers were calculated using variable-slope curve fitting (Excel). Strain PFEEC0089 was employed for the evaluation of immune sera from mouse studies. Combined $EC_{50}$ OPA titer data were plotted using GraphPad Prism to generate GMTs and associated $P$ values for statistical significance (Welch's unpaired $t$-test with log-transformed data).

Microbiology Spectrum

## ACKNOWLEDGMENTS

We'd like to thank the vaccine research and technology WGS team and Jonathan Lee for informatics support and various process development colleagues for long-chain O-antigen fermentation, purification, chemical conjugation, and analytic assays. For help with animal studies, we thank our comparative medicine colleagues. We also thank colleagues Ashlesh Murthy and Ognjen Sekulovic for their critical review of the manuscript.

## AUTHOR AFFILIATION

[1]Pfizer Vaccine Research and Development, Pearl River, New York, USA

## AUTHOR ORCIDs

Robert G. K. Donald  http://orcid.org/0000-0002-1002-954X

## FUNDING

| Funder | Grant(s) | Author(s) |
| --- | --- | --- |
| Pfizer Inc | | Robert G. K. Donald |
| | | Laurent Chorro |
| | | Duston Ndreu |
| | | Axay Patel |
| | | Srinivas Kodali |
| | | Zhenghui Li |
| | | David Keeney |
| | | Kaushik Dutta |
| | | Aniruddha Sasmal |
| | | Arthur Illenberger |
| | | C. Lynn Torres |
| | | Rosalind Pan |
| | | Natalie C. Silmon de Monerri |
| | | Ling Chu |
| | | Raphael Simon |
| | | Annaliesa S. Anderson |

## ADDITIONAL FILES

The following material is available online.

### Supplemental Material

**Supplemental material (Spectrum04213-23-S0001.pdf).** Tables S1 and S2; Fig. S1-S7.

### Open Peer Review

**PEER REVIEW HISTORY (review-history.pdf).** An accounting of the reviewer comments and feedback.

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
