## [Reviewer comments · Microbiology Spectrum]

Microbiology Spectrum

Preclinical validation of an *E. coli* O-antigen glycoconjugate for the prevention of serotype O1 invasive disease.

Robert Donald, Laurent Chorro, Duston Ndreu, Axay Patel, Srinivas Kodali, Zhenghui Li, David Keeney, Kaushik Dutta, Aniruddha Sasmal, Arthur Illenberger, C. Torres, Rosalind Pan, Natalie Silmon de Monerri, Ling Chu, Raphael Simon, and Annaliesa Anderson

Corresponding Author(s): Robert Donald, Pfizer Global Research and Development

Review Timeline:

Submission Date:	December 15, 2023
Editorial Decision:	March 12, 2024
Revision Received:	March 27, 2024
Accepted:	April 10, 2024

Editor: Fernando Navarro-Garcia

Reviewer(s): The reviewers have opted to remain anonymous.

Transaction Report:

DOI: <https://doi.org/10.1128/spectrum.04213-23>

Re: Spectrum04213-23 (Preclinical validation of an E. coli O-antigen glycoconjugate for the prevention of serotype O1 invasive disease.)

Dear Dr. Robert George Konrad Donald:

Thank you for the privilege of reviewing your work. Below you will find my comments, instructions from the Spectrum editorial office, and the reviewer comments.

Revision Guidelines

Sincerely,
Fernando Navarro-Garcia
Editor
Microbiology Spectrum

Reviewer #1 (Comments for the Author):

In the report, Chorro et al. evaluate a O1a CRM197 lattice glycoconjugate antigen in mice for efficacy against E.coli ST95 bloodstream infections. The O1 antigen was selected based on survey of >400 primary isolates. The O1a antigen from ST95, ST38, and ST59 were characterized at the structural level. The CRM197 conjugates were based on long chain O-polysaccharide and proved immunogenic in rabbits and mice and protective in the mouse challenge model. The sample sizes

are adequate and the statistical tests appropriate. The study is of high technical quality in all respects (genomics, NMR, glycoconjugate chemistry, vaccinology) and will be of interest to a broad readership. I have no major or minor critiques.

Reviewer #2 (Comments for the Author):

This study describes the development and evaluation of a conjugate vaccine candidate targeting the invasive *Escherichia coli* serotype O1, a pathogen of clinical relevance due to its association with bloodstream infections and its potential resistance to antimicrobial treatments. The vaccine candidate under investigation showed promising results in the animal studies, eliciting a strong IgG response and protecting mice against a lethal challenge with *E. coli* O1.

The manuscript is well-written, with no apparent spelling errors with generally accurate grammar. However, I recommend changing the orientation of all figures to a vertical presentation instead of horizontal and verifying the numbering of the tables for consistency.

To improve its clarity and overall impact, I propose the following modifications and corrections:

-Introduction

The introduction would benefit from a clearer statement regarding the broader significance of the research findings. Adding one or two sentences about the potential impact of these findings on public health and vaccine development could provide a strong closing to the introduction.

Line 68: Please specify the access numbers for these clinical trials. This detail will allow readers to access and review these trials directly.

- Results

Lines 111 - 113: For clearer interpretation, it is suggested to first mention that the phylogenetic tree was based on whole genome comparisons. Subsequently, the distribution of the STs within the tree could be outlined.

Lines 113 - 114: This section merits commencement in a new paragraph. Additionally, ensure the accuracy of reference 28, particularly in relation to *in silico* analysis and the specific strains mentioned.

Line 117: Antibiotic drug resistance... Introducing a new paragraph here will help in the interpretation of the results on antibiotic resistance.

Figure 2: Ensure that "*E. coli*" is consistently italicized.

Lines 159 - 165: The alignment of these sequences should be included in the supplementary information.

Line 169: Initiate a new subsection titled "Construction of the Conjugate Vaccine Candidate," for instance.

Lines 181 - 187: If these results are described but not showed, consider adding the relevant evidence to the supplementary material or removing the paragraph to maintain content integrity.

Lines 199 to 209: This reviewer is not familiar with serum opsonophagocytic assays (OPA) using the HL60 cell line and the data normalization method used by the authors. I recommend including references to studies where this methodology has been standardized and providing a more detailed explanation of it.

- Methods

Lines 373 - 385: This section omits details such as the number of animals per experiment, their living conditions, methods of blood collection or serum acquisition, bacterial dose inoculations, or statistical test for determining lethal dose 80 and survival analysis. Moreover, the approval status by an animal ethics committee is unclear. If applicable, please mention the protocol number and attach the approval certificate. An expanded description of this section is critical for completeness.

This study describes the development and evaluation of a conjugate vaccine candidate targeting the invasive *Escherichia coli* serotype O1, a pathogen of clinical relevance due to its association with bloodstream infections and its potential resistance to antimicrobial treatments. The vaccine candidate under investigation showed promising results in the animal studies, eliciting a strong IgG response and protecting mice against a lethal challenge with *E. coli* O1.

The manuscript is well-written, with no apparent spelling errors with generally accurate grammar. However, I recommend changing the orientation of all figures to a vertical presentation instead of horizontal and verifying the numbering of the tables for consistency.

To improve its clarity and overall impact, I propose the following modifications and corrections:

- Introduction

The introduction would benefit from a clearer statement regarding the broader significance of the research findings. Adding one or two sentences about the potential impact of these findings on public health and vaccine development could provide a strong closing to the introduction.

Line 68: Please specify the access numbers for these clinical trials. This detail will allow readers to access and review these trials directly.

- Results

Lines 111 - 113: For clearer interpretation, it is suggested to first mention that the phylogenetic tree was based on whole genome comparisons. Subsequently, the distribution of the STs within the tree could be outlined.

Lines 113 - 114: This section merits commencement in a new paragraph. Additionally, ensure the accuracy of reference 28, particularly in relation to in silico analysis and the specific strains mentioned.

Line 117: Antibiotic drug resistance... Introducing a new paragraph here will help in the interpretation of the results on antibiotic resistance.

Figure 2: Ensure that "*E. coli*" is consistently italicized.

Lines 159 – 165: The alignment of these sequences should be included in the supplementary information.

Line 169: Initiate a new subsection titled "Construction of the Conjugate Vaccine Candidate," for instance.

Lines 181 - 187: If these results are described but not showed, consider adding the relevant evidence to the supplementary material or removing the paragraph to maintain content integrity.

Lines 199 to 209: This reviewer is not familiar with serum opsonophagocytic assays (OPA) using the HL60 cell line and the data normalization method used by the authors. I recommend including references to studies where this methodology has been standardized and providing a more detailed explanation of it.

- Methods

Lines 373 – 385: This section omits details such as the number of animals per experiment, their living conditions, methods of blood collection or serum acquisition, bacterial dose inoculations, or statistical approaches for determining lethal dose 80 and survival analysis. Moreover, the approval status by an animal ethics committee is unclear. If applicable, please mention the protocol number and attach the approval certificate. An expanded description of this section is critical for completeness.

Reviewer #2 (Comments for the Author):

This study describes the development and evaluation of a conjugate vaccine candidate targeting the invasive Escherichia coli serotype O1, a pathogen of clinical relevance due to its association with bloodstream infections and its potential resistance to antimicrobial treatments. The vaccine candidate under investigation showed promising results in the animal studies, eliciting a strong IgG response and protecting mice against a lethal challenge with E. coli O1.

The manuscript is well-written, with no apparent spelling errors with generally accurate grammar. However, I recommend changing the orientation of all figures to a vertical presentation instead of horizontal and verifying the numbering of the tables for consistency.

Changed page format of all figures to standard (8" x 11")

To improve its clarity and overall impact, I propose the following modifications and corrections:

-Introduction

The introduction would benefit from a clearer statement regarding the broader significance of the research findings. Adding one or two sentences about the potential impact of these findings on public health and vaccine development could provide a strong closing to the introduction.

We expanded the last sentence to emphasize that we achieved proof-concept for an O-antigen based vaccine to prevent invasive disease caused by highly virulent encapsulated strains of this serotype. The discussion section also closes with broader implications in the context of our analogous serotype O25b data, which collectively show that O-antigens appear to be protective against invasive E. coli strains regardless of their potential for K-capsule expression *in vivo*.

Line 68: Please specify the access numbers for these clinical trials. This detail will allow readers to access and review these trials directly.

Added Janssen's ClinicalTrials.gov reference number (NCT04899336).

- Results

Lines 111 - 113: For clearer interpretation, it is suggested to first mention that the phylogenetic tree was based on whole genome comparisons. Subsequently, the distribution of the STs within the tree could be outlined.

A clarifying statement is included in line 108 to indicate that the phylogeny is based on whole genome comparisons.

*Lines 113 - 114: This section merits commencement in a new paragraph. Additionally, ensure the accuracy of reference 28, particularly in relation to *in silico* analysis and the specific strains mentioned.*

Initiated a new paragraph that also provides greater clarity for the basis of the *in silico* prediction of core type (28). "The presence of unique genes within the *waaQ* operon responsible for biosynthesis of each

outer core oligosaccharide can be used to predict LPS core type (ref 28). Accordingly, the presence of *waaV* was used to predict R1 core, *waaK* for R2, *waaD* for R3, *waaX* for R4 and *waaU* for K12....

Line 117: Antibiotic drug resistance... Introducing a new paragraph here will help in the interpretation of the results on antibiotic resistance.

New paragraph initiated as suggested.

Figure 2: Ensure that "E. coli" is consistently italicized.

Checked and confirmed.

Lines 159 - 165: The alignment of these sequences should be included in the supplementary information.

Added additional supplemental figure (S5) illustrating SNPs in O-antigen gene cluster alignments for the four ST648 strains mentioned (previously described in text as data not shown).

Line 169: Initiate a new subsection titled "Construction of the Conjugate Vaccine Candidate," for instance.

Added new heading.

Lines 181 - 187: If these results are described but not showed, consider adding the relevant evidence to the supplementary material or removing the paragraph to maintain content integrity.

Added additional supplemental figure (S6) showing LPS WB profiles and antisera cross-reactivity with *waaL* KO controls. This data was previously described in the text and in Table S1, without showing this raw data.

Lines 199 to 209: This reviewer is not familiar with serum opsonophagocytic assays (OPA) using the HL60 cell line and the data normalization method used by the authors. I recommend including references to studies where this methodology has been standardized and providing a more detailed explanation of it.

The assay is an adaptation of the classical killing-type OPA used for serological evaluation of pneumococcal vaccines. As recommended, we provide a citation which provides some general background (Romero-Steiner et al CVI 1997 4(4) p415-422). Our bactericidal OPA CFU data are plotted and interpolated without any normalization. For example, raw CFU counts for 11-point two-fold serum dilution titrations for two serotype O1a OPA assays are shown in Fig. 3. The methods section describes all essential details such as acceptance criteria for differentiated HL60 cells, the determination of assay specificity, internal assay controls, microplate staining procedure and data analysis.

- *Methods*

Lines 373 - 385: This section omits details such as the number of animals per experiment, their living conditions, methods of blood collection or serum acquisition, bacterial dose inoculations, or statistical test for determining lethal dose 80 and survival analysis. Moreover, the approval status by an animal

ethics committee is unclear. If applicable, please mention the protocol number and attach the approval certificate. An expanded description of this section is critical for completeness.

We expanded the Methods to include additional detail regarding dosing, blood draws and animal care. Also, we confirm that the animal procedures were approved by an IUCUC committee that includes qualified external reviewers and that the facility is accredited by the international Association for Assessment and Accreditation of Laboratory Animal Care (AAALAC).

Re: Spectrum04213-23R1 (Preclinical validation of an E. coli O-antigen glycoconjugate for the prevention of serotype O1 invasive disease.)

Dear Dr. Robert George Konrad Donald:

As a last suggestion, the authors are urged to improve the resolution of the figures presented because they are of low quality.

Your manuscript has been accepted, and I am forwarding it to the ASM production staff for publication. Your paper will first be checked to make sure all elements meet the technical requirements. ASM staff will contact you if anything needs to be revised before copyediting and production can begin. Otherwise, you will be notified when your proofs are ready to be viewed.

Sincerely,
Fernando Navarro-Garcia
Editor
Microbiology Spectrum

Reviewer #1 (Comments for the Author):

An excellent contribution.

Reviewer #2 (Comments for the Author):

The authors have addressed all comments made by this reviewer.

As a last suggestion, the authors are urged to improve the resolution of the figures presented because they are of low quality.